# What matters when doctors die: A qualitative study of family perspectives

**Martha A. Abshire**[1]*, **Marie T. Nolan**[1], **Sydney M. Dy**[2], **Joseph J. Gallo**[1,3]

**1** Johns Hopkins University School of Nursing, Baltimore, Maryland, United States of America, **2** Department of Health Policy and Management, Bloomberg School of Public Health, Johns Hopkins University, Baltimore, Maryland, United States of America, **3** Department of Mental Health, Bloomberg School of Public Health, Johns Hopkins University, Baltimore, Maryland, United States of America

* mabshir1@jhu.edu

## Abstract

### Background

The challenges of supporting the end-of-life preferences of patients and their families have often been attributed to poor understanding of the patient's condition. Understanding how physicians, as patients, communicate their end-of-life care preferences to their families may inform shared decision making at end of life.

### Objectives

The purpose of this study was to understand what matters to families of physicians when decision making with and for a physician who is approaching the end of life.

### Design

Cross-sectional qualitative design.

### Participants

We conducted interviews with family members of deceased physicians.

### Approach

We analyzed the data using the constant comparison method to identify themes.

### Key results

Family members (N = 26) rarely were unclear about the treatment preferences of physicians who died. Three overarching themes emerged about what matters most to physicians' families: (1) honoring preferences for the context of end-of-life care; (2) supporting the patient's control and dignity in care; and, (3) developing a shared understanding of preferences. Families struggled to make decisions and provide the care needed by the physicians at the end of life, often encountering significant challenges from the healthcare system.

**Data Availability Statement:** All relevant data are within the paper and its Supporting Information files.

**Funding:** JJG and MTN were co-PIs on R01 NR014068 funded by the National Institute of Nursing Research. The funder had no role in study design, data collection and analysis, decision to publish, or preparation of the manuscript.

**Competing interests:** The authors have declared that no competing interests exist.

## Conclusions

Even when disease and prognosis are well understood as in this group of physicians, families still experienced difficulties in end-of-life decision making. These findings highlight the need to specifically address preferences for caregiver, care setting and symptom management in shared end-of-life decision making conversations with patients and families.

## Background

Families often need to face the challenge of understanding and respecting loved ones' end-of-life preferences while managing care through declining health. From the point of view of families in the United States, quality of end-of-life care decreased from 2000 to 2013 despite the many efforts to improve services provided at the end of life [1]. Globally, quality end of life care is a public health concern with wide variability in hospital use of life-sustaining treatments [2–4]. Common barriers to quality end-of-life care include unrealistic expectations of the patient for cure, clinician concern about taking away hope, and unrealistic patient expectations for recovery [5]. Poor quality of care at end of life impacts symptom control, spiritual needs and emotional processing of patients and families, putting families at risk for prolonged grief [6,7].

Although highly individualized, a systematic review on the concept of 'a good death' found that patients and their families value maintaining autonomy, control over symptoms, opportunities for closure or reconciliation and limiting the sense of 'being a burden' [8,9]. Some have described a good death as a dignified death, but for many, the circumstances surrounding the end of life illness, communication among family members and the health care team and even a basic understanding that the patient is dying prevent providing a path to dignity for the dying patient [10,11] While many intersecting dynamics influence the complexity of end-of-life decision making, one common thread is the understanding of the disease, prognosis and consequences of the illness, which may influence poor quality of end of life care in populations with low health literacy [10,12,13].

Physicians are an informative group to study because they have knowledge of prognosis and treatment options that many lay persons lack [14]. Without these barriers, physicians and their family members may be able to focus on other aspects of quality of life when making decisions about care at the end of life. Although there has been limited writing about physician end-of-life decision making, one study of physician's widows highlighted the importance of financial planning and estate planning and another study found that physicians report a high rate of documenting an advanced directive, but still are reluctant to discuss their preferences for end-of-life treatment with their family and physicians [15,16]. Therefore, the family members of deceased physicians have a unique perspective which may inform our understanding of end-of-life decision making for others. In essence, we wanted to know, "What matters most to their families when physicians die?" The purpose of this study was to identify how family members interpreted physician values and supported decision making at the physician's end of life.

### Design

The Precursors Study is an observational cohort study of medical students and graduates of Johns Hopkins School of Medicine between 1946 and 1964. The Precursors Study was

originally designed to determine whether there are psychological, social and behavioral characteristics associated with morbidity and mortality, particularly from heart disease [17,18]. All 1,337 students who matriculated into the graduating classes of 1948 to 1964 of the Johns Hopkins University School of Medicine participated. As this cohort advanced in age, end-of-life issues became more important; a mixed methods study was designed to examine physician preferences for end-of-life care and their approach to decision making. As part of that study, to understand what decisions were made as the physician was dying, we conducted after-death interviews with family members of older physician participants to elucidate their perspectives. For the analysis described in this paper, we focused on how family members described what mattered to them in supporting end-of-life decision making. All study procedures including informed consent for this study were reviewed and approved by the Johns Hopkins University School of Medicine Institutional Review Board (IRB00051731).

## Sample selection

As previously reported, in annual surveys physicians were asked to state their wishes regarding the use of ten medical interventions in the event of irreversible brain injury based on the medical directive (CPR, mechanical ventilation, intravenous fluids, surgically placed feeding tube for nutrition, dialysis, chemotherapy, major surgery, invasive diagnostic tests, blood or blood products, and antibiotics) [19]. Physicians were categorized into three classes of end-of-life preferences based on the latent class model: "most aggressive," "intermediate care," and "least aggressive." We then used the classes as a sampling frame to select family members of decedent physicians for interview in order to have all three categories represented. For each physician, a single family member was contacted with one voice representing the family experience. All participants were contacted first by letter with an option to opt out of a phone call related to the present study. Using a random order stratified by preference class, physicians or the next of kin provided to the Precursors Study were contacted by phone. Verbal informed consent was provided by each participant prior to beginning the interview.

## Interview strategy

Interviews were recorded using an audio recorder by study team members experienced with semi-structured interviews (a nurse with research experience and a post-doctoral student with a background in anthropology). We used a semi-structured interview guide that contained the following broad topics regarding end-of-life decision making: (1) the extent to which the physician-patient had expressed his/her wishes; (2) how decisions were made; (3) the extent to which care was consistent with values and preferences; (4) the family member's experience with making decisions at the end of life for anyone else; (5) whether care seemed different because the patient was a physician; and (6) regrets regarding the care and treatment of the physician patient. Trustworthiness of our data was enhanced at the data collection phase by regular debriefing of the interviewers to encourage standardized administration of the questions [20]. Interviews lasted approximately 20 minutes. Audio recordings were transcribed for coding.

## Analytic strategy

Coding of transcripts involved sorting the data into large-level categories arrived at through group consensus in team discussion (broad coding). A working codebook was developed in order to reach consensus on emerging codes and address the need for additional codes. Team members (JG, MN and MA) independently coded transcripts, and suggested large subject areas (e.g., "what is most important at the end of life") that were present in the interview and

noted text that was not covered by existing codes. To ensure trustworthiness, we reviewed the transcripts as a team and discussed any discrepancies to reach consensus, entering the final coding in nVIVO (version 12, QSR International). We then created a definition for each code to ensure integrity of application across interviews. We strove for coding categories generated from meaning inherent in the data rather than pre-specified categories. We then re-read broad code reports and discussed them as a group, fine coding the text within the broad codes. The constant comparative method, which allows analysts to move iteratively between codes and text to derive themes, guided the identification of themes [21]. Data saturation was discussed and recruitment ended when the team agreed that no new perspectives were surfacing through interviews.

## Results

### Study sample

In all, we conducted interviews with 26 family members after the death of the physician (Table 1). Most physician-patients were male and of advanced age at the time of death. Family members who participated in interviews were predominantly women and about half of the family members were spouses.

### What matters to families

Our themes are described below with supporting quotes. Additional quotes supporting themes are included in S1 Appendix. As described elsewhere, the cohort had varied attitudes towards aggressive treatment and over time, those who changed had a less aggressive attitude toward treatment [19].

**Table 1. Characteristics of deceased physician-patients and their family members (N = 26 Deceased Physician patients and 26 Family Members).**

| Characteristic | |
|---|---|
| Physician Characteristics | |
| Average Age of physician at death | 84.4 years |
| Sex of deceased physician (n, %) | |
| Male | 26 (100%) |
| EOL treatment preference groups (2005–2011), (n, %) | |
| Most Aggressive | 6 (23%) |
| Changing attitudes towards aggressiveness | 11 (42%) |
| Least Aggressive or Moderate | 9 (35%) |
| Average year of death (range) | 2013 (2005 to 2016) |
| Family Member Characteristics | |
| Sex of family member, (n, %) | |
| Male | 6 (23%) |
| Female | 20 (77%) |
| Relationship to physician, (n, %) | |
| Wife | 14 (54%) |
| Daughter | 6 (23%) |
| Son | 5 (19%) |
| Friend | 1 (4%) |

## Honoring preferences for the context of end-of-life care

Families commonly needed to address the specific physician preference regarding *who* would provide care and *where* the care would take place. Several family members talked about expecting that the physician-patients would exert control over their preferred treatments but families did not expect the conflicts that occurred when determining who would provide care (family vs a hired person) and where the care would be received (home, the hospital, nursing home).

> I mean, he loved having me there, too (daughter), but it was her (wife)—she was the one that he wanted to be taking care of him . . . "Well, is it okay if she (formal paid caregiver) keeps coming?" And he says, "I want your mother to take care of me." And so he kept going, "I want Mom, to take care of me." But he knew that she couldn't do that full-time.
>
> Family member 3265

Physicians were concerned about 'being a burden' to their families. To mitigate the physician's sense of being a burden, some families were able to divide tasks and labor among other family members. Family members described the importance of having multiple people available to support the physician-patient and, when possible, the caregiver at the end of life. Family members saw their role as both advocating for the physician-patient values and preferences and also as maintaining the cohesion of other family members by checking in with other family members on the decisions being made.

> He elected to include my brother and me and we were happy to help him. It was just not something he wanted to do by himself. And we were all on the same page, for sure. And we had good communication about it and we were all able to reach what we felt like was the best decision we could make at the time.
>
> Family member 8365

Family members described good collaborative relationships with the medical team caring, indicating an additional benefit because the physician-patients were known and/or respected in the medical community in which they were being treated. However, family members sometimes pitted themselves against the medical team or the health care system in their struggle to balance the physician-patient's preferences and changing medical needs. One family member recalled inhumane treatment during one of the physician-patient's many hospitalizations and said that the family had to demand a basic level of respect. "You're kind of a lump to be moved around and dealt with [without family to advocate for you]." Families who had these negative experiences with the healthcare system felt a sense of betrayal given the physician-patients' identity as a physican.

## Supporting the patient to preserve control and dignity in care

An essential dilemma for family members was how to respect the dignity of the physician-patients as families provided care at the end of life. Many of the physicians had very prestigious and accomplished medical and research careers, and a strong identity as one entrusted to provide complex medical care to others. Several patients and family members perceived a deeply distressing loss of dignity in the transition from physician to patient following the onset of diminished physical and/or mental capacity.

> You know, like I say, once he starts wandering around the halls naked in the middle of the morning, we don't have—<laughs> we didn't have much of a ground to stand on. . .I

mean, he was always more or less presentable, but, you know, he always also had crumbs on his pants when we showed up there and, you know, just minor things. But then he didn't notice, so why should we complain. . .We defended his independence and his freedom as long as we could until it was obvious that he just couldn't do it any more.

Family member 4871

Physician-patients had a very clear understanding of how they wanted their symptoms to be managed. Based on their own medical practice many physicians were able to make expert decisions for themselves balancing treatments and symptom burden at the end of life. Some family members recalled the importance of balancing of pain management and lucidity among physician-patients.

What he said was he knows that it kind of took you out of your—you weren't aware of what was going on around you per se, and he wanted to be aware of it, and so even though he was in great pain he would rather have his wits about him than to be not in pain and not understand what was going on.

Family member 8755

It was incredibly important to families to honor the late-life pursuits of the physician-patients. For several physician-patients, creating a legacy of their life's work was an important factor contributing to sense of self or dignity. Some chose to prepare their own obituaries as a way to summarize the accomplishments that had the greatest meaning for them. Others sought to bring a life-long program of research to closure with the assistance of colleagues. Several physician-patients maintained a sense of dignity and purpose by setting goals which were supported by their family members.

And when [he] was in decline, he realized that he had mountains of data that had not yet, ah, gone into print. . .And this younger man was kind enough to work with him, when [he] couldn't do it himself and get the data prepared for publication. . . .And he finally found a place for that paper. So all of [his] work is now—well that one is not in print yet, but it will be—will be—will all be in print, and for this I am very grateful.

Family member 5118

### Developing a shared understanding of preferences

A shared understanding of end-of-life care preferences was highly valued, however many family members reported developing a shared understanding was difficult due to the indirect way that physicians communicated their preferences. Physician-patients described their preferences for end-of-life care casually throughout their careers, after encounters with different types of patients in their practice or in different healthcare settings such as nursing homes. Very few family members remembered a time when the physician family member intentionally and carefully explained preferences for care at end of life, although most physicians had documented advance directives for treatment preferences.

They had conversations and we'd have other relatives and when their older relatives died and all the stuff they said and how they dealt with them. And comments my dad had said about, "Well, that person, there's no quality of life there. There's nothing to do about that,"

things like that. It was pretty clear to us where their mindset was. . .It always boils down to a grey area and a judgement call. So it's—you're talking about a loved one's life so it's always a little bit of "I hope I'm doing the right thing," but we felt we were doing the right thing.

Family member 9415

When families had a shared faith or belief about the meaning of life and death, decisions seemed to proceed more easily. This shared understanding of the value for life or quality of life, particularly was helpful in making decisions that were not specified by advance directives. To facilitate these discussions, some physician's families found their own resources about meaning in life extremely valuable.

We also read the Atul Gawande's book "Being Mortal" which was a lovely book, but "The Conversation" I have told many friends about it because it takes off the personal. I mean it sort of makes it sort of third person. The third person is asking you these questions and you're filling them in. And so, it was just like filling in a questionnaire.

Family member 5868

Of course, not all family members shared the same values or beliefs about what types of treatments ought to be provided to maintain life and what treatments could be declined based on patient preferences.

My sister is a very huge pro-lifer person, and my parents were very concerned that she would be keeping them going when they don't want to keep going, so that's when they actually changed me to be their person to make decisions, because they were fearful that my eldest sister, who was at that point going to be in charge of everything—they were afraid that she would not follow their choices, so . . .

Family member 8495

Even with a shared understanding of care and treatment preferences, sometimes it was impossible to provide the level of care needed in the way the physician-patient preferred, which often left family having to make difficult decisions.

So she absolutely, positively refused to allow hospice to be any part of her care. So the place that she was at, at the end of her life, finally said, "You know, you really need to involve hospice," and at that point, again, she wasn't really conscious anymore and so, I said, "You're right, I do."

Family member 9536

## Discussion

In this study we explored how family members of deceased physicians understood the end-of-life preferences of their loved ones. In answer to the question, "What is important to their families when doctors die?" we identified 3 themes: 1) Honoring preferences for the context of end-of-life care, 2) Supporting the patient to preserve control and dignity in care, and 3) Developing a shared understanding. Specifically, families identified the importance of honoring the physicians' preferences for who would care for them and where they would receive care. To advocate for these preferences, families divided tasks among family members and

united to offset the impersonal aspects of the healthcare setting. Several families prioritized respecting the dignity of the physician-patients by respecting their preferences regarding symptom management, other goals of care and legacy-making. Finally, physician family members felt empowered when they worked together to understand the physician's care preferences regarding difficult decisions made before death and this collaboration seemed to bring an element of peace to the family when reviewing their decisions after the physician's death.

Before discussing the implications of our findings, the limitations should be discussed. Although the Precursors data have provided important evidence, it is a unique cohort of predominantly white, male physicians. Importantly, their family members may also benefit from higher than average health literacy, financial means, and as they noted, receive different health care than those without family connections to healthcare providers. However, we believe if even physicians and their families had difficulties with decision making and with providing essential elements of care at the end of life despite a very high number with advance directives, higher education and adequate financial resources, that lessons learned from their experiences may highlight important gaps in the ways end-of-life decision making is approached. Many types of end-of-life experiences and care preferences may not have been discussed. Despite our attempts to maintain trustworthiness and transparency in our methods, it is possible that our own biases may have influenced how we interpreted interviews and future studies will need to verify and validate this research. Also, family members who agreed to continue to be involved in the Precursors Study may be more likely to share the values of the physician participant or have had less conflict than those who did not participate Finally, while we refer to the 'family' throughout the manuscript, we only interviewed one family member, which may not completely reflect the family as a group.

One strength of our study is that we purposively sampled physicians with a range of attitudes towards aggressive treatment and regardless of specific treatment decisions physicians expressed preferences for *where* they wanted to spend their last days and *who* they preferred to provide that care. Another strength was that none of the family members reported that the physician-patient had difficulty understanding prognosis or treatment options, allowing us the rare opportunity to explore additional care preferences that were challenging to address, even for the families of physicians. Although more than 80% of physicians in Precursors had advance directives, many family members still expressed the challenge of making more subtle end-of-life treatment and care decisions [16]. Many of the care preferences identified by physicians would fit into existing models for goals of care conversations, such as the Serious Illness Conversation Guide, which typically broadly cover goals, fears, trade-offs and family considerations [22]. However the physician preferences may highlight the need to specifically address certain key areas such as preferences for caregiver, care setting and symptom management. Conversations may also be balanced with consideration for practical needs, financial considerations and the well-being of the caregiver [23].

One criterion for quality of end-of-life care is whether the patient died at home given that across studies and cultures, many patients with life-limiting illness prefer to die at home rather than in a hospital [24]. Some studies have demonstrated that physicians are less likely to die in the hospital than others [25], however a recent study found no difference between physicians and non-physicians in location of death [26]. Physicians were, however, more likely to receive palliative care than non-physicians. The conflict experienced by many family members in this study about whether or not they could adequately provide the home care desired by the patient is not unique to physician families [23]. Families may be unprepared to care for dying loved ones at home because most formal advance care planning documents focus on treatment preferences rather than the context of the care (including the location of the care and the preferred caregivers). A systematic review of studies that explored how patients with life-threatening

illness define a "good death" included these and many other contextual factors such as having spiritual support, having a sense of completion, and an opportunity to help others right up to the end of life [27]. Asking patients with life-threatening illness how they would define a good death may reveal contextual issues that can be planned for just as families plan care consistent with treatment preferences. These conversations may provide a deeper understanding of the patient's values and may create a shared sense of understanding among patients and families.

While quality of care at end of life is understood to depend on many factors that are not treatment oriented, the role of dignity as a foundational concept underpinning decision making remains underappreciated when planning for end-of-life care [28]. For physicians who are members of one of the most respected of professions, dignity and the loss of control were at stake as they approached their own deaths. Their illnesses threatened their self-value and their perceptions of their value to others, key components of attributed, extrinsic dignity [29]. Our findings were highly aligned with the 3 main components of the Chochinov Dignity Model: 1) illness-related concerns (such as independence through functional and cognitive decline); 2) dignity-conservation (through activities such as creating legacy, maintaining autonomy and goal setting); and, 3) social dignity (which involves the role of social relationships to influence feelings of worth) [30,31].

Physician and ethicist, Edmund Pellegrino defined intrinsic human dignity as the inherent worth in all human beings and extrinsic dignity as the worth that humans assign to themselves or others which can be significantly diminished by illness and disability (Pellegrino, 2008) [32]. In a literature review of "dying with dignity," authors confirmed this duality of intrinsic worth and extrinsic qualities such as autonomy, meaningfulness and interpersonal connection [29]. In reflecting on what family members told us about the death of their physician relative, we were struck by the salience of the role of dignity in all aspects of treatment and care decisions. The legacy of physician-patients was a source of accomplishment, admiration and pride in nearly all family member interviews.

Family members seemed committed to protecting the intrinsic dignity of the physican-patient against what seemed to be diminished extrinsic dignity assigned to the physician-patient by various health providers or dehumanizing practices in the health care system. Dignity-conserving strategies have been demonstrated to improve the quality of outcomes at end of life [28,30,31]. Opportunities to promote dignity suggested by this study include controlling symptoms and medication side effects, recognizing intrinsic dignity despite cognitive or physical decline and establishing goals and a legacy with the patient and their family as a sign of respect for the patient as a person.

Family members spoke with admiration of the careful planning of many of the physicians, including financial and funeral planning as well as preparation of formal advance directive documents or living wills. It seemed that much of this work was done by the physicians to be proactive despite the uncertainty of end of life and prevent creating additional burden for the family. Even so, families were often left to muddle through the implications of advance directives and casual conversations over the years to guide decisions because many end-of-life decisions were not specified by the advance directives.

Although the physician-patients were more informed than most lay patients about their illness, prognosis and treatment options and their family caregivers were also well educated, family caregivers reported concerns that have been reported among lay family caregivers. For example, family caregivers talked about their desire to reach consensus with other family members on treatment and care decisions for the physician patient but recognized that some level of conflict among the family members might occur [33]. Similar to other studies of caregivers of persons with serious illness, faith beliefs and religious practices provided great comfort to some of the family members of the physician-patients especially near death, but

## Box 1. Recommendations for focusing on what matters in end-of-life conversations

Discuss what creates a sense of meaning, motivation or purpose for the patient

Establish the preferred context of care including who will provide care and where care will be provided

Focus on symptom control preferences regarding pain, level of consciousness, breathlessness and physical function.

Consider how preferences may vary in a scenario with functional versus cognitive impairment

Discuss how family and friends can form a social support team, dividing tasks and responsibilities in the face of increased caregiving needs

Address treatment and preferences in the context of understanding the intrinsic dignity of the patient, supporting the patient's feelings of worth through attributed dignity

divergent beliefs created conflict [34]. While current US practice is to communicate with a single proxy decision maker, family structures and dynamics rarely are so clear. Improving shared decision making is a national healthcare priority, essential to which is considering the family structure, especially among racially diverse groups [35, 36]. Without this shared understanding, surrogate decision makers draw from a variety of sources to make decisions about care and treatment such as past conversations and their own values, but often feel uncertain if their decisions reflect patient preferences [37].

Our findings have implications for end-of-life conversations. What matters to the families of physicians when the physician is dying and after his/her death should directly challenge healthcare providers to be more clear in conversations about their own end-of-life preferences. Suggestions for clinical practice are provided in Box 1, based on our themes and findings. The generous willingness of families to discuss painful memories of challenging decision making should also be honored through changes in our clinical practices. If we don't understand our own priorities in end-of-life decision making as healthcare providers, how can we hope to support others in their process, especially those with less medical knowledge?

## Supporting information

**S1 Appendix. What Matters to their Families When Doctors Die?.** Additional quotes from interviews of family members of physicians in the Johns Hopkins Precursors Study presented by theme.
(DOCX)

**S1 Checklist. COREQ (COnsolidated criteria for REporting Qualitative research) checklist.**
(PDF)

## Acknowledgments

1. We gratefully acknowledge the contributions of the family members who shared their difficult grief journeys and opened themselves to engage in this important conversation.

2. We have not presented this work at a conference.

## Author Contributions

**Conceptualization:** Martha A. Abshire, Marie T. Nolan, Sydney M. Dy, Joseph J. Gallo.

**Formal analysis:** Martha A. Abshire, Marie T. Nolan, Joseph J. Gallo.

**Funding acquisition:** Marie T. Nolan, Joseph J. Gallo.

**Methodology:** Martha A. Abshire, Marie T. Nolan, Joseph J. Gallo.

**Supervision:** Marie T. Nolan, Sydney M. Dy, Joseph J. Gallo.

**Writing – original draft:** Martha A. Abshire.

**Writing – review & editing:** Marie T. Nolan, Sydney M. Dy, Joseph J. Gallo.

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
