## [Decision Letter · Decision Letter 0]

19 Mar 2020

PONE-D-20-04418

What matters to families when doctors die: "keeping life normal, having a plan and then help at the end"

PLOS ONE

Dear Dr Abshire,

Thank you for submitting your manuscript to PLOS ONE. After careful consideration, we feel that it has merit but does not fully meet PLOS ONE’s publication criteria as it currently stands. Therefore, we invite you to submit a revised version of the manuscript that addresses the points raised during the review process.

We would appreciate receiving your revised manuscript by May 03 2020 11:59PM. To enhance the reproducibility of your results, we recommend that if applicable you deposit your laboratory protocols in protocols.io, where a protocol can be assigned its own identifier (DOI) such that it can be cited independently in the future. For instructions see: http://journals.plos.org/plosone/s/submission-guidelines#loc-laboratory-protocols

We look forward to receiving your revised manuscript.

Kind regards,

Edison I.O. Vidal, MD, MPH, PhD

Academic Editor

PLOS ONE

Journal Requirements:

2. Please modify the title to ensure that it is meeting PLOS’ guidelines (https://journals.plos.org/plosone/s/submission-guidelines#loc-title). In particular, the title should be "specific, descriptive, concise, and comprehensible to readers outside the field" and in this case the title may not convey sufficient description of the methods or analysis undertaken.

Additional Editor Comments (if provided):

The reviewers have raised some important questions and made some useful suggestions on how your paper could be improved.

Both reviewers made important comments regarding the focus of the manuscript. I concur with the reviewers as you have provided readers with conflicting messages regarding the aims of this study. On P. 4, lines 74-77, you stated that “In essence, we wanted to know, “What matters most to their families when physicians die?” The purpose of this study was to identify common challenges and strategies used by families in end-of-life decision making in a situation where the patient, a physician, had an excellent understanding of treatment options and prognosis.” However, on P. 5, lines 88-89, it was stated that “present study was designed to examine physician preferences for end-of-life care and their approach to decision making”. Please, state clearly all aims of the study and confirm that analyses are consistent with those aims throughout the manuscript.

Although you have stated that “Study procedures including informed consent are annually reviewed and approved by the Johns Hopkins University School of  Medicine Institutional Review Board (IRB00051731)”, that description by itself does not guarantee that the subjects that were interviewed provided informed consent.

Both reviewers requested more information about patients and interviewees. Please consider presenting a table with the characteristics of patients (e.g. gender, age [mean and standard deviation or median and interquartile range], main diagnostic leading to the patient’s death, medical specialty, classification of end-of-life preferences) and another table with the characteristics of the family members who were interviewed (e.g. gender, age, profession, type of relationship with the patient, whether the interviewee was the primary healthcare representative of the patient or not).

Finally, I noticed that several items of the COREQ (Consolidated criteria for REporting Qualitative research) (http://intqhc.oxfordjournals.org/content/19/6/349.long) were not addressed in this report. For instance, there is no information about how participants were selected in terms of inclusion and exclusion criteria. Please, revise the manuscript accordingly and provide a COREQ checklist indicating the page numbers where each item was addressed as supplementary material. This is a very important issue that must be addressed by the authors since adherence to appropriate reporting guidelines is one of the publication criteria adopted by PLOS ONE (https://journals.plos.org/plosone/s/criteria-for-publication).

Reviewers' comments:

Reviewer's Responses to Questions

**Comments to the Author**

1. Is the manuscript technically sound, and do the data support the conclusions?

Reviewer #1: Partly

Reviewer #2: Yes

2. Has the statistical analysis been performed appropriately and rigorously? 

Reviewer #1: N/A

Reviewer #2: N/A

3. Have the authors made all data underlying the findings in their manuscript fully available?

Reviewer #1: Yes

Reviewer #2: Yes

4. Is the manuscript presented in an intelligible fashion and written in standard English?

Reviewer #1: Yes

Reviewer #2: Yes

5. Review Comments to the Author

Reviewer #1: This is a well written qualitative study exploring what matters to family of patient-physicians at the end-of-life. Please consider my comments/suggestions below to improve the quality of the paper.

Introduction

- "From the point of view of families, quality of end-of-life care decreased from 2000 to 2013 despite the many efforts to improve services provided at the end of life." Where was this assessed? As this is an international journal, it would be adequate that such affirmations were contextualised in more detail as end-of-life care quality varies hugely among and within countries.

- "Poor quality of care at end of life impacts families even after the death of a loved one, putting them at risk for prolonged grief." I would like to see information about the importance of good quality end-of-life care for the people who are at the end of their lives too or about the public health need for improvement in end-of-life care more broadly.

- "Many studies have identified that patients often have a poor understanding of prognosis and treatment options preventing high quality end of life care." This sentence sounds dubious - please rewrite this for clarity.

Design

- "As this cohort advanced in age, end-of-life issues became more important; the present study was designed to examine physician preferences for end-of-life care and their approach to decision making." I understand the study was about the families’ views about what matters at the end-of-life of patient-physicians, and not explore the patient-physicians’ preferences? Please clarify this.

- I would like to see more details about the methodology used and rationale for it: I understand constant comparison to be more of an inductive qualitative technique or method for data analysis, rather than a study methodology in itself. Also, normally, constant comparison method is jointly used with theoretical sampling, which was not the case in this particular study. Please provide more information/justification on these aspects.

- I think it would be important for the reader to contextualise the findings if the diagnoses of the patient-physicians were provided (cancer; dementia; etc).

Results

- Page 8: "As described elsewhere, the cohort had varied attitudes towards aggressive treatment, despite their advanced age." - why would one expect attitudes towards end-of-life treatment vary with age?

- Appendix:

I think the quotes within the themes “Challenges with the healthcare system” and “Control over symptoms” do not quite represent its titles and therefore these need careful revision. For example, the first quote in “Challenges with the healthcare system” seems to be about the particular benefit of being a physician when it comes to end-of-life preferences and care as there is better communication and respect of patient preferences. Within “Control over symptoms”, the first sentence clearly seems to be about acceptance; and the second and third quotes which seem to be more about having control over one’s overall life and decisions, rather than control over symptoms.

Discussion

Typos: lines 333 and 351

Line 345: “life-threatening illness” – do you mean life-limiting illness?

Reviewer #2: Thank you for submitting your article on what matters to families when doctors die. It is a well-crafted article and a worthwhile read. Some additional revisions, however, are necessary to further improve the manuscript. I hope you find the comments helpful.

My main comment is that of article focus. While the title and part of the manuscript place the focus on informal care giver experience, both the topics for the semi-structured interview and majority of your results focus on what mattered to the now-deceased practitioners - even if by proxy. Caregiver own needs seem less of a focus in the way you present the results, too. Nonetheless, I appreciate that post-bereavement interviews with informal carers are likely to involve carer focus on the needs of the patient even if the interview questions emphasise carer needs and experiences. Literature on conceptualising carers and service users/patients as dyads, which acknowledges the complex interdependencies of need may be useful here. Overall, the focus of the manuscript should be clearer and consistent throughout - whichever way you chose to approach this.

I understand that the wordcount is not limitless, but currently the background section is rather brief. Clearer situating of your research in wider literature on the topic would be beneficial. Studies specifically on healthcare professional own death and end of life care are not abundant, but they do exist (e.g. Gray CH. How will your wife cope when you die? Doctors' widows supply some answers. Canadian Medical Association Journal. 1980 Jan 26;122(2):206.). What matters to families and care givers is covered fairly extensively and again can help you situate your study. This would in turn help to further clarify to the reader why your research matters. You are right in saying that a physician-patient cohort have considerably more healthcare literacy and understanding of illness trajectories than other patient groups, and that therefore they are better-able to make informed advance decisions - but tell the reader why this matters? Is it, for example, about looking at what issues are more universal across patient populations? Is it about greater caregiver well-being, if carers are spared some complex decision making? You do provide a purpose for your study, but explaining 'why' that is the purpose would help your reader.

In the discussion section, your manuscript would benefit from a clearer flow of the overall argument. At the moment the discussion is slightly disjointed and the many implications you mention could be connected in a more coherent manner. This may be because - as mentioned earlier - the focus shifts between informal caregivers and physicians at the end of life... as well as between recommendations for practice and specific focus on dignity. What is the one main point you would like the reader to take away from this article? What is the main contribution that you are making to the body of literature on the topic? A clearer structure, directionally building to that main point, would help the reader follow the arguments.

Below are some smaller points:

- Although your interviews touched on divergence of family member wishes and approaches, overall you spoke to (primary) informal caregivers. I am therefore wondering if reference to family wishes and priorities (i.e. implying the whole family) is somewhat misleading.

- You acknowledge from the start that physicians are a 'unique' patient population, but so are their informal caregivers. You touch on this in the discussion, but could also benefit from briefly discussing this when defining the sample. Caregivers, too, may have a greater than average health literacy, be practicing or retired healthcare staff themselves, be more affluent than the general population and afford better care, etc. Some of this may also be important in the limitations section.

- PLoS One has an international readership and it would be helpful if you specified the context when relevant (e.g. lines 61-62 refer to the US). Similarly, some country-specific terminology - such as 'Medical Directive' - would benefit from a very brief definition or addition of alternative terms used elsewhere (e.g. 'living will').

- It would be interesting to know the proportions of participants who were caring for physicians with 'most aggressive', 'intermediate' and 'least aggressive' care preferences.

- In the participant table you break down adult children by gender, but do not do so for spouses. A uniform approach in either direction is advisable.

- It is more conventional to describe study participants in the methods section (e.g. see Christenson A, Johansson E, Reynisdottir S, Torgerson J, Hemmingsson E. Women's perceived reasons for their excessive postpartum weight retention: a qualitative interview study. PLoS One. 2016;11(12)). Similarly, adding participant codes or pseudonyms to the quotes you use in the result section is both conventional and helps the reader understand if you are over-relying on any one transcript or if quotes used reflect all/most of your participants.

- Line 321 - it could be clearer whom you are referring to by using the term 'providers'

- Lines 338-339 - please add a reference.

Once again, thank you for your submission and I hope you find the comments helpful.

6. PLOS authors have the option to publish the peer review history of their article (what does this mean?). If published, this will include your full peer review and any attached files.

Reviewer #1: No

Reviewer #2: Yes: Dr Rasa Mikelyte

---

## [Author Response · Author response to Decision Letter 0]

3 Jun 2020

Please see attached file for response to the editor and reviewers.

---

## [Editor Report · Decision Letter 1]

10 Jun 2020

What matters to families when doctors die:  a qualitative study of family perspectives

PONE-D-20-04418R1

Dear Dr. Abshire,

We’re pleased to inform you that your manuscript has been judged scientifically suitable for publication and will be formally accepted for publication once it meets all outstanding technical requirements.

Kind regards,

Edison I.O. Vidal, MD, MPH, PhD

Academic Editor

PLOS ONE
---

## [Editor Report · Acceptance letter]

12 Jun 2020

PONE-D-20-04418R1 

What matters when doctors die: A qualitative study of family perspectives 

Dear Dr. Abshire:

I'm pleased to inform you that your manuscript has been deemed suitable for publication in PLOS ONE. Congratulations! Your manuscript is now with our production department. 

Kind regards, 

on behalf of

Professor Edison I.O. Vidal 

Academic Editor

PLOS ONE